**Data Availability Statement:** All relevant data are within the manuscript and its Supporting Information files.

**Funding:** Financial support for this research was provided by the Florida Fish and Wildlife

# Testing the efficacy of lionfish traps in the northern Gulf of Mexico

**Holden E. Harris** [1,2]*, **Alexander Q. Fogg**[3], **Stephen R. Gittings**[4], **Robert N. M. Ahrens**[2], **Micheal S. Allen**[2,5], **William F. Patterson III**[2]

**1** School of Natural Resources and Environment, Institute of Food and Agriculture Sciences, University of Florida, Gainesville, Florida, United States of America, **2** Department of Fisheries and Aquatic Sciences, School of Forest Resources and Conservation, Institute of Food and Agriculture Sciences, University of Florida, Gainesville, Florida, United States of America, **3** Okaloosa County Board of County Commissioners, Destin-Fort Walton Beach, Florida, United States of America, **4** Office of National Marine Sanctuaries, National Oceanic and Atmospheric Administration, Silver Spring, Maryland, United States of America, **5** Nature Coast Biological Station, Institute of Food and Agriculture Sciences, University of Florida, Cedar Key, Florida, United States of America

* holdenharris@ufl.edu

## Abstract

Spearfishing is currently the primary approach for removing invasive lionfish *(Pterois volitans/miles)* to mitigate their impacts on western Atlantic marine ecosystems, but a substantial portion of lionfish spawning biomass is beyond the depth limits of SCUBA divers. Innovative technologies may offer a means to target deepwater populations and allow for the development of a lionfish trap fishery, but the removal efficiency and potential environmental impacts of lionfish traps have not been evaluated. We tested a collapsible, non-containment trap (the 'Gittings trap') near artificial reefs in the northern Gulf of Mexico. A total of 327 lionfish and 28 native fish (four were species protected with regulations) recruited (i.e., were observed within the trap footprint at the time of retrieval) to traps during 82 trap sets, catching 144 lionfish and 29 native fish (one more than recruited, indicating detection error). Lionfish recruitment was highest for single (versus paired) traps deployed <15 m from reefs with a 1-day soak time, for which mean lionfish and native fish recruitment per trap were approximately 5 and 0.1, respectively. Lionfish from traps were an average of 19 mm or 62 grams larger than those caught spearfishing. Community impacts from Gittings traps appeared minimal given that recruitment rates were >10X higher for lionfish than native fishes and that traps did not move on the bottom during two major storm events, although further testing will be necessary to test trap movement with surface floats. Additional research should also focus on design and operational modifications to improve Gittings trap deployment success (68% successfully opened on the seabed) and reduce lionfish escapement (56% escaped from traps upon retrieval). While removal efficiency for lionfish demonstrated by traps (12–24%) was far below that of spearfishing, Gittings traps appear suitable for future development and testing on deepwater natural reefs, which constitute >90% of the region's reef habitat.

Conservation Commission (Grant No. 13416 to R. N. M. Ahrens and H. E. Harris). Support for H. E. Harris was provided by the National Science Foundation Graduate Research Fellowship Program (Grant Nos. DGE-1315138 and DGE-182473). The funders had no role in study design, data collection and analysis, decision to publish, or preparation of the manuscript. Opinions, findings, or conclusions expressed in this document do not necessarily reflect the views of our supporting organizations.

**Competing interests:** The authors have declared that no competing interests exist.

## Introduction

Invasive Indo-Pacific lionfish (*Pterois volitans/miles* complex, hereafter "lionfish") are now well established in the western Atlantic, including the Caribbean Sea and Gulf of Mexico [1], and have recently invaded the Mediterranean Sea [2]. Lionfish occupy a wide diversity of invaded marine habitats, including coral reefs, subtropical artificial and natural reefs [3], seagrass beds [4], mangroves [5], estuaries [6], mesophotic reefs [7–9], and upper continental slope reefs [10]. High population densities of lionfish [3,11] have caused reductions in native reef fish abundances [12,13], altered marine communities [14,15], and likely exacerbate current stressors on marine systems [16,17]. As invasive lionfish populations do not appear to be controlled by native predators [18–20], reducing lionfish biomass is a top priority for marine resource managers [21,22]. Population and ecosystem models predict that high levels of fishing mortality over a broad geographic range will be necessary to control lionfish populations on a regional scale [15,23–25].

Lionfish are primarily removed by spearfishing on SCUBA [21,26]; however, invasive lionfish have been observed at depths >300 m [10] and diver removals are generally limited to depths <40 m. Over 557,000 km$^2$ of benthic habitat in the western Atlantic invaded range of lionfish lies within mesophotic and upper-bathyal depths of 40–300 m [27,28]. Although survey capacity for deepwater reefs is relatively limited [28], lionfish density has been documented to be higher on mesophotic reefs than on corresponding shallower reefs [9,29–31]. Deepwater lionfish populations likely disrupt food webs on mesophotic reefs [7] and provide refuge for larger and more fecund individuals [32]. These protected source populations can provide larvae for sink regions [25,33] and undermine shallow-water control efforts [8,32]. Innovative harvest technologies have been proposed for deepwater lionfish removals, including modifications to existing spiny lobster traps [34], weaponized remotely operated vehicles [22,35], and novel trap designs [36]. Such technologies may offer a safe method for lionfish removal from deepwater reefs inaccessible to spearfishers and could allow a single vessel to multiply effort via simultaneous gear deployments.

Collaborative work by members of non-profit organizations, Florida Fish and Wildlife Conservation Commission, and the US National Oceanic and Atmospheric Administration have resulted in lionfish trap prototypes [37], which have been further developed into a non-containment trap model [38]. The collapsible, non-containment 'Gittings trap' is designed to allow lionfish and native fish to freely move over the trap's footprint, with traps closing during retrieval (**Table 1**, **Fig 1**). Gittings traps are made from common and inexpensive materials,

**Table 1. Goals of the Gittings lionfish trap with design attributes employed to achieve those goals.**

| Goals | Trap characteristics |
|---|---|
| Attract and capture lionfish | Plastic lattice provides vertical structure to attract lionfish from nearby habitats. |
| Limit bycatch | Non-containment design prevents fish mortality prior to retrieval. Lack of bait reduces recruitment of non-targeted species. Presence of lionfish may also deter native fishes. |
| Prevent ghost fishing | Non-containment design with downward-opening curtain minimizes the likelihood of continued mortality ("ghost fishing") if a trap is lost. |
| Limit habitat damage | Collapsed trap falls quickly through the water to facilitate placement accuracy. Traps are placed on sand and low relief habitats where snagging is less likely. Low center of gravity reduces likelihood of movement. |
| Allow for easy transport on fishing boats | Traps are collapsible and stackable. |
| Allow for safe release of bycatch | Able to recompress bycatch by descending in a closed trap, releasing fish upon contact with the bottom. |

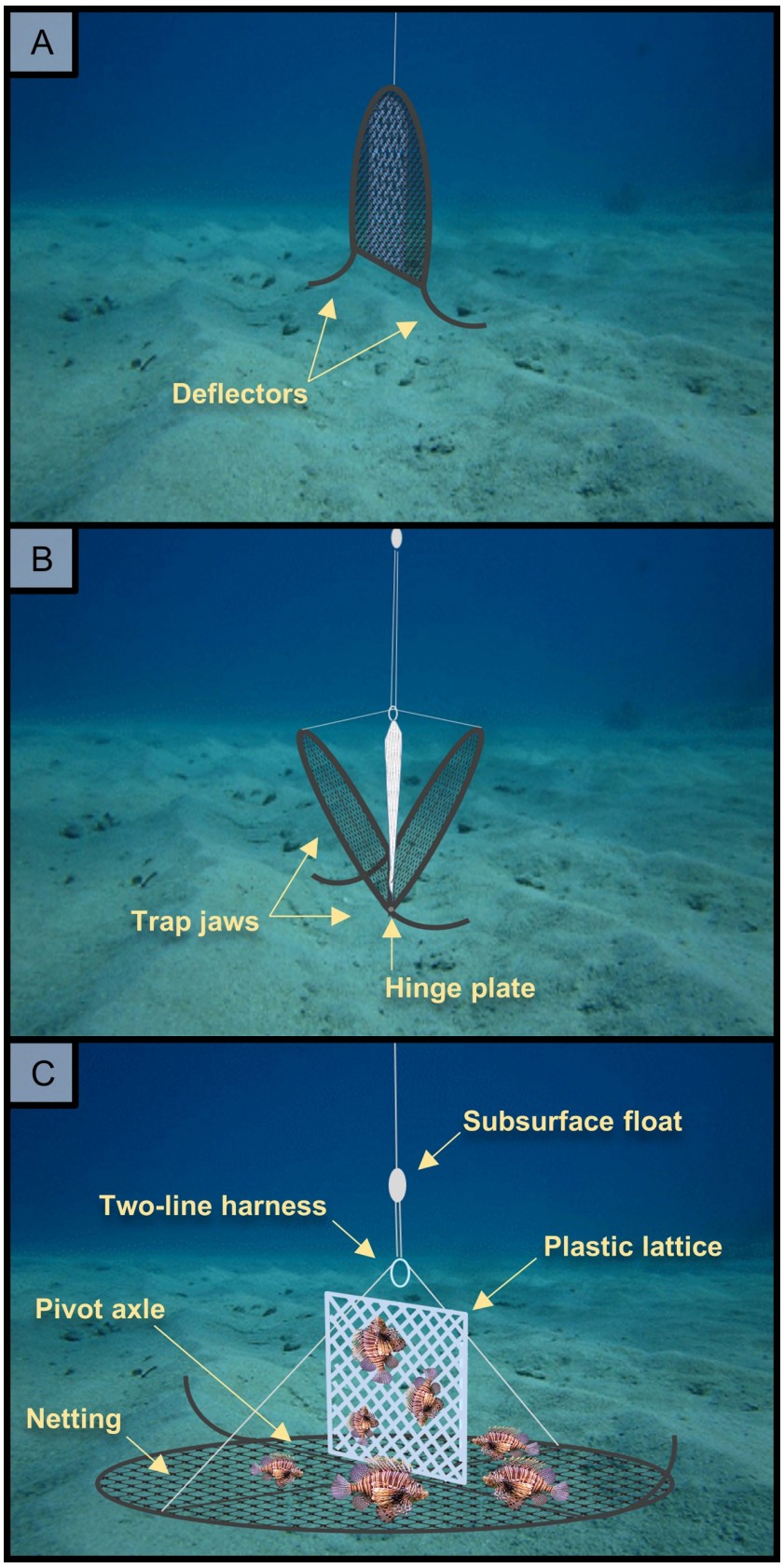

**Fig 1. Schematic of Gittings lionfish trap deployment.** Traps are designed to A) descend closed and B) open when the curved deflectors contact the seafloor. C) The traps remain open during deployment then close when the trap is ascended during retrieval.

allowing for construction in remote locations where specialized materials may be difficult to source. A low cost of production could expand harvesting capacity for the nascent lionfish commercial fishery, which is currently constrained by inconsistent supply [39,40]. Such a commercial deepwater lionfish fishery may offer additional livelihood strategies for fishers and improve coastal food security [41]. However, it is necessary to further evaluate a potential new harvest gear for possible undesirable effects–including bycatch, habitat damage, entanglement, and ghost fishing–before it is permitted and widely implemented. Traditional fish traps have a broad catch composition [42] and their widespread use has contributed to severe overfishing on many Caribbean coral reef systems [42,43]. Given their potential for bycatch and overfishing, moratoriums on fish traps have been in place in US Atlantic and Gulf of Mexico waters for decades [44], with the exception of a limited trap fishery for Atlantic black sea bass (*Centropristis striata*) [45].

Here, we report results from testing Gittings traps near artificial reefs in the northern Gulf of Mexico (nGOM). Our objectives were to 1) assess gear performance of Gittings traps and 2) examine how lionfish and native fish recruitment (number of fish observed within the trap footprint at the time of retrieval) and catches (fish landed aboard the vessel from a trap) were affected by the lionfish density on the adjacent artificial reef and different Gittings trap deployment configurations. Different trap configurations tested strategies for changing the soak time, the number of Gittings traps deployed, and their distance from the adjacent artificial reef. Gear testing Gittings trap performance involved evaluating deployment success (% of traps that successfully opened on the seabed), lionfish escapement (% of individuals that escaped traps upon retrieval), and whether traps moved on the seabed while deployed. Lionfish size distributions were also compared between Gittings trap catches by distance to the artificial reef and *in situ* size distributions obtained from spearfishing catches. We consider how the findings from this study can inform further research and development of lionfish traps and innovative harvest technologies to control deepwater lionfish populations.

## Methods

Twelve Gittings traps were constructed in May and June 2018. Gittings traps have hinged jaws that allow for the trap to remain closed and travel vertically through the water column (**Fig 1A**), then open when the curved deflectors contact the seafloor (**Fig 1B**). Trap jaws were made from 4.5 m long sections of #6 rebar (19 mm diameter) bent into two half-hoops with a curved extension on one end of each jaw to act as deflectors for opening the trap when it contacts the seafloor (**Fig 2**). The jaws pivot around a 2 m long center axle made with #6 round bar. The axle and jaws are connected with two metal hinge plates (4 cm x 10 cm) each with holes approximately 20 mm in diameter. Trap netting consisted of 3 $m^2$ of 22 mm diameter mesh nylon netting (#420 green knotless). A sheet of plastic lattice (71 cm x 75 cm with 2.5 cm openings) provided vertical structure for attracting lionfish (**Figs 1C and 2**). A two-line harness was attached to the apex of each trap jaw using 12-strand Dyneema fiber rope (Amsteel Blue, 2.78 mm diameter). To prevent the harness line from fouling within the trap, an inline syntactic foam float was secured at the apex of the harness. An instructional video for building similar Gittings traps is provided at https://youtu.be/ta8WInxyXFA.

Gittings traps were deployed in depths of 33–37 m near eight artificial reefs on the nGOM Florida shelf. These included four poultry transport units (i.e., chicken coops), one steel

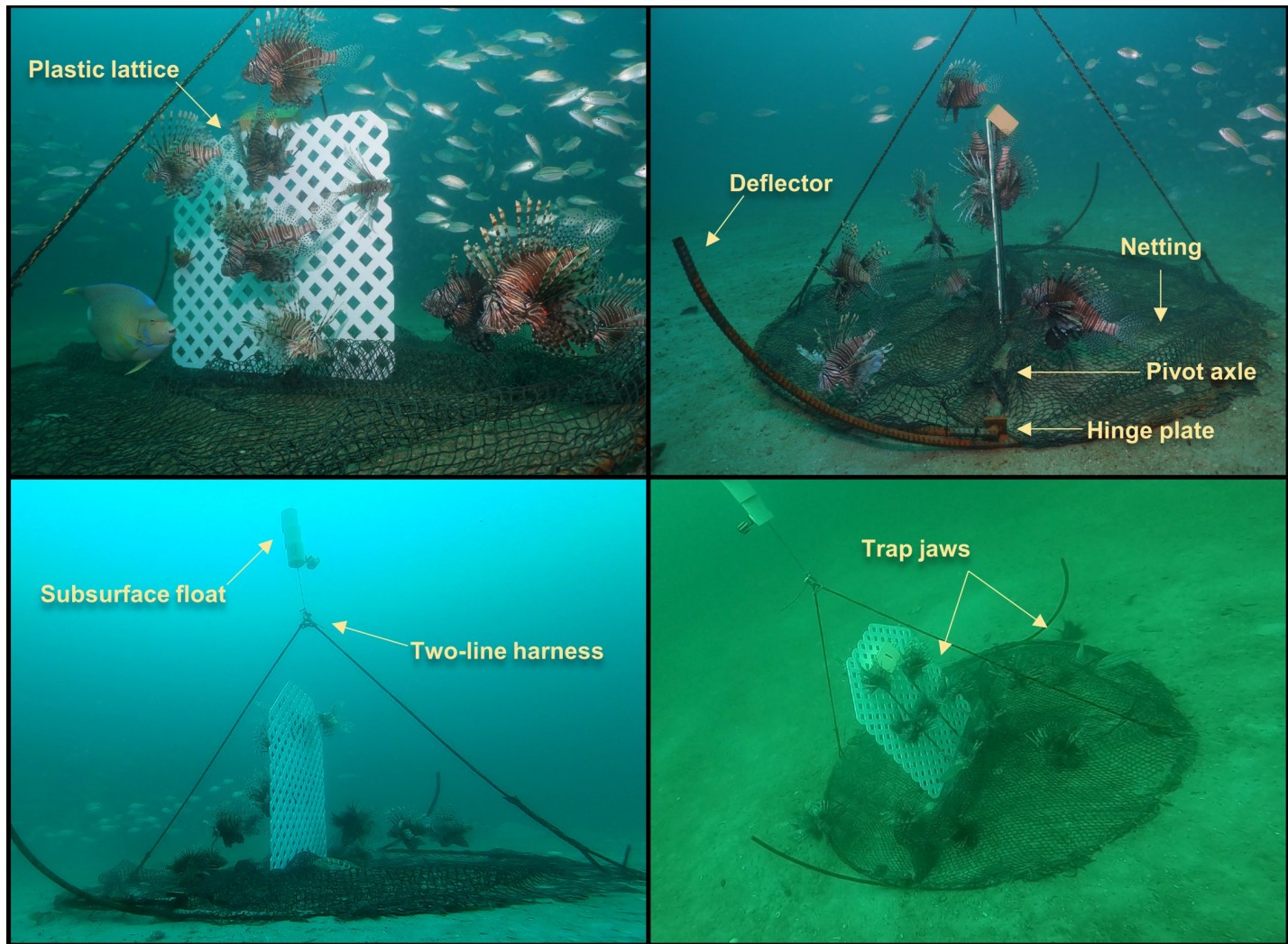

**Fig 2. In water photos of a Gittings lionfish traps deployed near artificial reefs in the northern Gulf of Mexico.** Lionfish are attracted to the structure made from plastic lattice. The trap jaws are constructed from rebar and bent to make deflectors that open the trap when it contacts the seafloor. The jaws open around a central pivot axle connected with a hinge plate. Fish are captured in the mesh nylon netting when the jaws are lifted via a two-line harness. Images: A. Fogg and H. Harris.

pyramid, one cement mixer, and two military tanks located approximately 30 km south of Destin, Florida (**Fig 3**). The artificial reefs were approximately 3 m x 2 m x 2 m (length x width x height) in size and deployed on sand bottom. No other known reef habitats (artificial or natural) were within 300 m of a given artificial reef. Lionfish density at each reef was surveyed immediately prior to trap deployment and trap retrieval. Surveys were conducted with point-counts by divers on SCUBA within a 15-m wide cylinder with the artificial reef at the center [46]. The diver survey included a lionfish count on the opposite sides of the artificial reef, followed by a count of lionfish within the reef structure [46,47]. Lionfish density (fish per 100 m$^2$) was computed as abundance (number counted) divided by the area sampled (177 m$^2$).

Trap deployments and retrievals (n = 58 replicates; **Table 2**) were conducted during June through December 2018. Deployment factors for number of traps (two levels: single or paired), distance to reef [three levels: near (5 m), intermediate (15 m), or far (65 m)], and soak time (categorical with five levels: 0.25 day, 1 day, 4–5 days, 8 days, or 12–14 days) were randomized for each deployment. Traps were deployed from the vessel and allowed to descend freely.

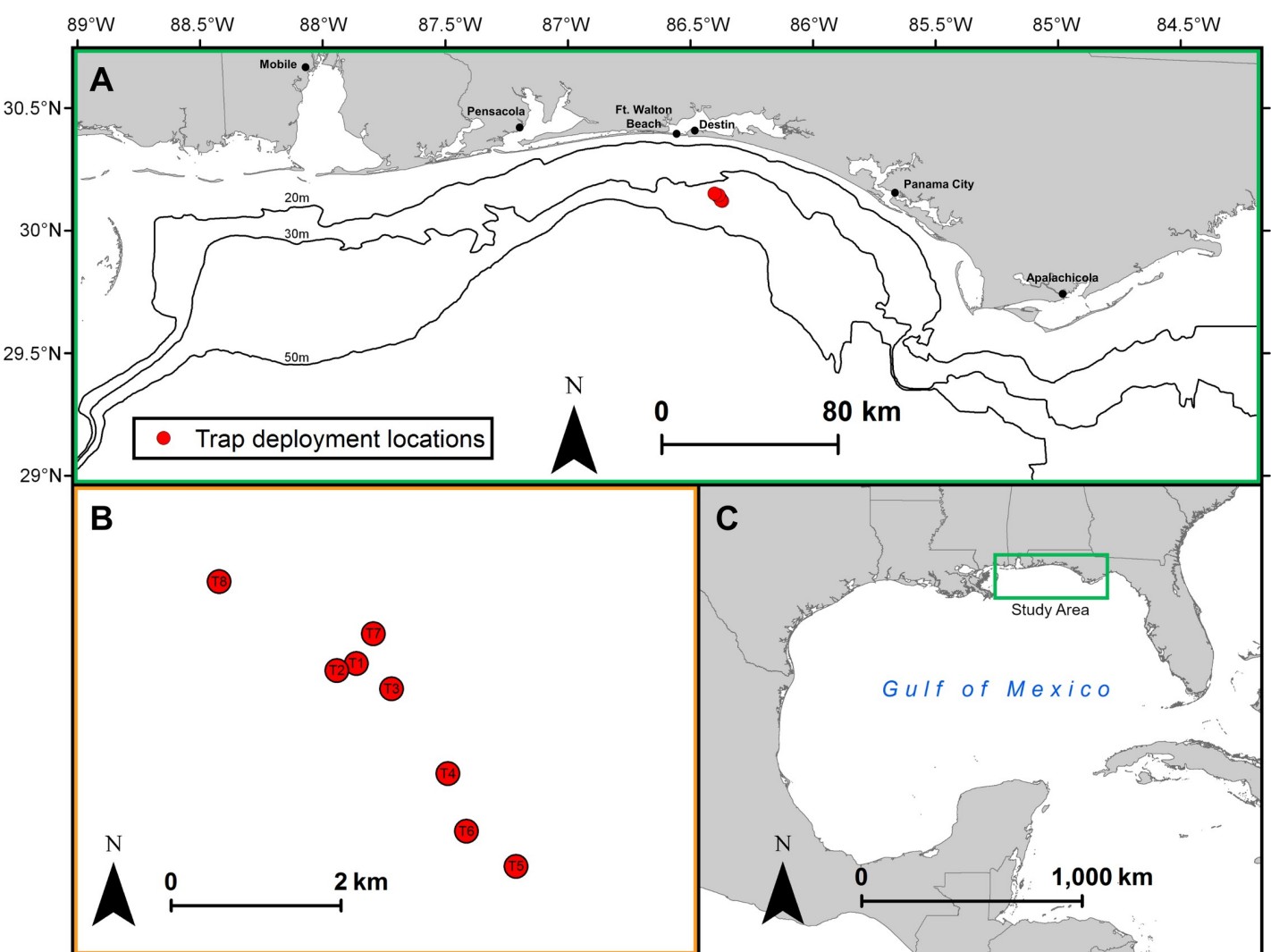

**Fig 3. Study site locations for testing Gittings lionfish traps.** A) Traps were deployed adjacent to artificial reefs approximately 30 km offshore NW Florida in depths of 33–37 m. The eight reef study sites were B) separated by >300 m and C) located in the northern Gulf of Mexico.

Single-trap treatments included one trap deployed at the reef. Paired-trap treatments included two traps deployed simultaneously ~3 m apart. Deployment success (i.e., a trap landed upright and opened) for each trap deployment (n = 82 total traps deployed including paired treatments) was noted by SCUBA divers during the survey and unsuccessful deployments were corrected. Underwater visibility ranged from 5–12 m, thus near traps were within direct visual range of the reefs by SCUBA divers, intermediate distanced traps were sometimes within visual range or just outside, and far traps were outside of visual range. Traps were retrieved by SCUBA divers using two, 22-kg lift bags filled with air (video of trap retrieval provided at https://youtu.be/Tf8K6ZwQV_Y). Recruitment of lionfish and native fish (i.e., the number of fish within the trap footprint) was documented by a SCUBA diver prior to retrieval and subsequently compared to the catch. Although we attempted to use time-lapse camera units to document higher resolution recruitment to the traps, the camera failure rates during this study were too high for the data to be useful.

**Table 2. Replicates by deployment factor tested.**

| Soak time / | Distance to reef | | |
|---|---|---|---|
| # traps | **5 m** | **15 m** | **65 m** |
| **0.25 days** | | | |
| Single | 5 | 4 | 0 |
| Paired | 0 | 0 | 0 |
| **1 day** | | | |
| Single | 4 | 4 | 0 |
| Paired | 0 | 0 | 0 |
| **4–5 days** | | | |
| Single | 5 | 4 | 1 |
| Paired | 3 | 3 | 1 |
| **8 days** | | | |
| Single | 2 | 0 | 2 |
| Paired | 2 | 0 | 2 |
| **12–14 days** | | | |
| Single | 4 | 2 | 2 |
| Paired | 4 | 2 | 2 |

Number of Gittings trap replicates by deployment configuration with factors of soak time (0.25 day, 1 day, 4–5 days, 8 days, or 12–14 days), number of traps deployed (single or paired), and distance to artificial reef (5 m, 15 m, or 65 m).

Generalized linear mixed models (GLMMs) were computed to test the effect of deployment factors (i.e., number of traps, distance to artificial reef, and soak time) on 1) lionfish recruitment, 2) lionfish catch, 3) native fish recruitment, and 4) native fish catch (**Eq** 1). Lionfish density on the adjacent artificial reef site was included as a covariate. Individual artificial reef sites were sampled multiple times with different deployment configurations and soak times, thus the reef site was included in the GLMMs as a random effect (random intercept) and assumed to be normally distributed with a mean of zero and variance $\sigma^2$. Deployment factors and the lionfish density covariate were evaluated at an experiment-wise error rate ($\alpha$) of 0.05. Quantile-quantile (QQ) plots were used to determine if errors were best fit with a normal, lognormal, Poisson, or negative binomial distribution. Likelihood was estimated with Laplace approximation based on GLMM fitting and inference protocols [48]. Analyses were conducted in R (version 3.6.1) using the LME4 [49] and MASS [50] packages. See supplemental material for R code and raw data. The QQ plots can be produced by running the R code.

$$\text{Recruitment}_{\text{Lionfish}} \sim \text{Negative binomial} (\mu)$$

$$\text{Catch}_{\text{Lionfish}} \sim \text{Negative binomial} (\mu)$$

$$\text{Recruitment}_{\text{Native fish}} \sim \text{Negative binomial} (\mu)$$

$$\text{Catch}_{\text{Native fish}} \sim \text{Negative binomial} (\mu)$$

$$\text{Reef} \sim \text{N}(0, \ \sigma^2)$$

$$\log(\mu) = \textit{Trap number} + \textit{Distance} + \textit{Soak time} + \textit{Lionfish density} + (1|\textit{Reef site})$$

Eq 1

Total length (TL) was measured for trap-caught lionfish (n = 163) to 1) compare size of lionfish caught in the trap and distance from reef and 2) compare sizes of trap-caught lionfish to those caught via spearfishing (n = 3,063) during the same time period from similar artificial reefs in the study region. Lionfish size samples via spearfishing were collected during monthly

lionfish culls by trained volunteer divers [51]. Spearfishing collections were subset for lionfish captured from artificial reefs during the same time period as the trap testing (June–December 2018). During spearfishing, divers attempted to capture all lionfish observed on a given artificial reef regardless of lionfish size. Lionfish detectability was assumed to be unbiased for larger individuals given the relatively low structural complexity of the artificial reefs and the fact that previous serial removals on similar nGOM artificial reef habitats produced homogeneous size distributions [46]. Nonparametric two-sample Kolmogorov–Smirnov (KS) tests were used to compare TL distributions given that the distributions were non-normal with multiple modes present. The TL data met the assumptions of the KS test; data were independent, ordinal, uncensored, ungrouped, and followed a continuous distribution. The KS tests compared the difference between TL distributions 1) for trap-caught lionfish by distance to reef (5 m, 15 m, and 65 m), and 2) between trap-caught or spear-caught lionfish. Given that lionfish recruitment on subtropical reefs varies by season [51,52], TL distributions from trap-caught lionfish was also compared between seasons with a KS test. Seasons were defined as late summer (June–September) and early winter (November–December). Differences in mean size by weight (in grams) were calculated with the weight-length relationship $W = a \times TL^b$ using allometric parameters $a = 3.09E\text{-}6$ and $b = 3.27$ estimated for nGOM lionfish [53].

Spearfishers were informed and consented to information about their catch being used for research purposes. Lionfish collection by researchers followed humane sampling protocol with euthanasia via pithing the brain case as reviewed and approved by the University of Florida's Institutional Animal Care and Use Committee (UF IACUC Protocol #201810225). Authorized use of Gittings traps in US federal waters to collect lionfish for scientific research was granted by a Letter of Acknowledgment from the US National Marine Fisheries Service Southeast Regional Office in accordance with the definitions and guidance at 50 CFR 600.10.

## Results

### Gear testing

Traps deployed upright and opened during 56 of 82 (68%) deployments (video of Gittings trap opening during deployment is provided at https://youtu.be/XlyNuLxEqgQ). Traps landed on sand and there was no entanglement on habitat features or attached organisms. The potential for Gittings trap movement was tested during two severe weather events. On Sept 4–5, 2018 the center of Tropical Storm Gordon passed ~150 km west of 12 deployed traps with maximum sustained winds of >110 km/h and recorded seas >5 m. Traps were retrieved two days later with all 12 found upright and no change in location, although traps were heavily fouled with algae (video of trap retrieval after Tropical Storm Gordon is provided at https://youtu.be/7wZpe5fOozs). Then, on Oct 9–10, 2018 Category 5 Hurricane Michael passed ~100 km east of six deployed traps with maximum sustained winds >250 km/h and seas >15 m. Traps could not be retrieved for over a month due to extensive damage in the region but, upon retrieval, all six traps were upright at their deployment locations. While these observations indicate high-energy storm events did not move Gittings traps on the seafloor, it is currently unclear if, or to what extent, movement would have occurred with surface buoys attached to the traps.

Gittings traps that were successfully deployed recruited lionfish from nearby artificial reefs (**Fig 2**). However, an issue of lionfish escapement was clearly indicated with 56% of the lionfish that had recruited to traps escaping during retrieval. The proportion of lionfish that escaped did not appear to correlate with higher recruitment of lionfish to traps (see R code for scatter plots). Because traps were closed for ascent by lift bags that divers attached to traps and filled with air, closing took 3–5 seconds and divers often observed lionfish swimming out in this

time. Native fish were not observed escaping during trap closing, although detection of native fishes was an apparent issue, as described below.

### Lionfish and native fish trap recruitment

A total of 327 lionfish recruited to Gittings traps during 82 trap sets (n = 58 deployments, including paired sets) with 141 lionfish caught (**Fig 4, left**). Trap bycatch of native fish consisted of 28 individuals recruiting to the traps and 29 individuals caught from nine different species (**Fig 4, right**). The fact that one more native reef fish was caught (i.e., landed on the boat from the trap retrieval) than documented as recruited by divers prior to retrieval indicates an apparent detection error by the divers. Four native fishes captured in traps were regulated species: two scamp (*Mycteroperca phenax*), one Gulf flounder (*Paralichthys albigutta*), and one blue angelfish (*Holacanthus bermudensis*). Additional native fish catches consisted of 15 sand perch (*Diplectrum formosum)*, four tomtate grunt (*Haemulon aurolineatum*), two bank sea bass (*Centropristis ocyurus*), two porgies (*Calamus* spp.), one soapfish (*Rypticus* spp.), and one polka-dot batfish (*Ogcocephalus radiatus*). Native fish TL measurements were similar to lionfish with a range of 151–350 mm.

Lionfish recruitment ranged from 0 to 20 per trap and lionfish catch ranged from 0 to 14 per trap (video of a Gittings trap with high recruitment is provided at https://youtu.be/1vzByPMm7hQ). QQ plots indicated that the error structures for the recruitment and catch models (Eq 1) were best fit with a negative binomial distribution, which is typical for count

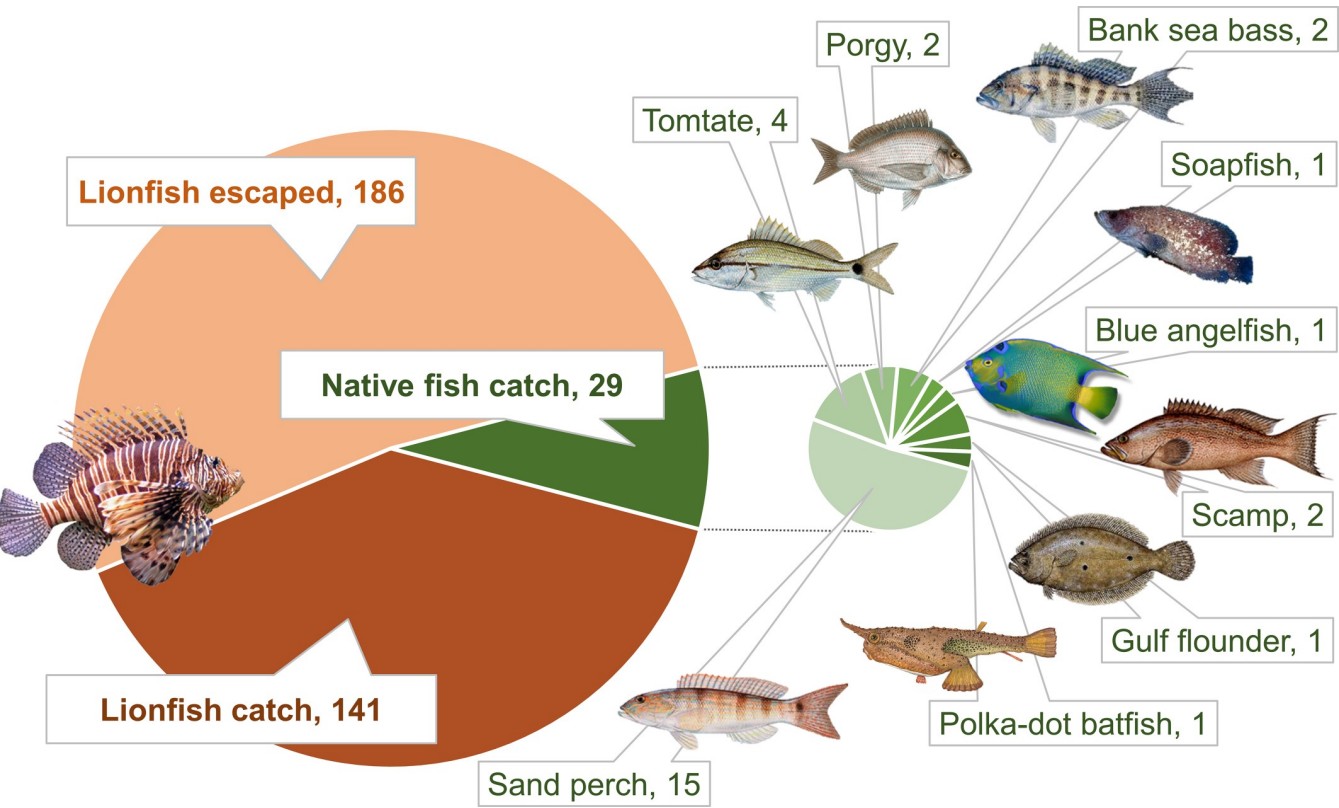

**Fig 4. Number and species of fishes in the Gittings traps.** Total counts of lionfish caught, lionfish escaped, and native fish caught in traps deployed near northern Gulf of Mexico artificial reefs during 82 trap deployments. Sizes of pie slices correspond to the proportion of total fish caught. Fish images are not drawn to scale.

**Table 3. Generalized mixed model results testing deployment factors on lionfish and native fish species recruitment and catches by Gittings traps.**

| Model | Factor | Level | Odds ratio | Parameter estimate (#fish / trap) | 95% CI | z | P* |
|---|---|---|---|---|---|---|---|
| Lionfish recruitment | (Intercept) | | | 2.77 | 1.22–6.33 | 6.33 | **0.015** |
| | Trap number | Paired | 0.56 | 1.55 | 0.64–3.74 | 2.09 | 0.195 |
| | Distance | 15 m | 0.95 | 2.63 | 1.55–4.46 | 4.23 | 0.860 |
| | Distance | 65 m | 0.03 | 0.08 | 0.00–0.69 | 0.02 | **0.001** |
| | Soak time | 1 day | 1.89 | 5.24 | 2.83–9.64 | 18.2 | **0.042** |
| | Soak time | 4–5 days | 1.82 | 5.04 | 2.60–9.78 | 17.8 | 0.078 |
| | Soak time | 8 days | 1.66 | 4.60 | 2.05–10.4 | 17.2 | 0.217 |
| | Soak time | 12–14 days | 1.50 | 4.16 | 2.11–8.17 | 12.3 | 0.245 |
| | LF density | | 1.02 | 2.83 | 2.74–2.94 | 3.00 | 0.152 |
| Lionfish catch | (Intercept) | | | 4.38 | 1.64–11.7 | 11.7 | **0.003** |
| | Trap number | Paired | 0.52 | 2.28 | 1.09–4.73 | 2.46 | 0.080 |
| | Distance | 15 m | 0.76 | 3.33 | 1.71–6.48 | 4.93 | 0.419 |
| | Distance | 65 m | 0.06 | 0.26 | 0.04–2.19 | 0.13 | **0.009** |
| | Soak time | 4–5 days | 1.09 | 4.77 | 2.15–10.6 | 11.6 | 0.837 |
| | LF density | 8 days | 0.31 | 1.36 | 0.35–4.95 | 1.54 | 0.076 |
| | Soak time | 12–14 days | 0.16 | 0.70 | 0.22–2.10 | 0.34 | **0.001** |
| | LF density | | 1.02 | 4.47 | 4.29–4.64 | 4.74 | 0.331 |
| Native fish recruitment | (Intercept) | | | 0.26 | 0.06–1.13 | 1.13 | 0.073 |
| | Trap number | Paired | 0.23 | 0.06 | 0.02–0.20 | 0.05 | **0.018** |
| | Distance | 15 m | 2.05 | 0.53 | 0.22–1.28 | 2.62 | 0.107 |
| | Distance | 65 m | 0.00 | 0.00 | NA | NA | NA |
| | Soak time | 1 day | 0.36 | 0.09 | 0.01–0.91 | 0.32 | 0.383 |
| | Soak time | 4–5 days | 4.73 | 1.23 | 0.35–4.27 | 20.2 | **0.014** |
| | Soak time | 8 days | 2.47 | 0.64 | 0.08–5.04 | 12.4 | 0.389 |
| | Soak time | 12–14 days | 1.62 | 0.42 | 0.08–2.12 | 3.43 | 0.559 |
| | LF density | | 1.00 | 0.26 | 0.25–0.27 | 0.27 | 0.866 |
| Native fish catch | (Intercept) | | | 0.23 | 0.03–1.95 | 1.95 | 0.178 |
| | Trap number | Paired | 0.26 | 0.06 | 0.02–0.19 | 0.05 | **0.024** |
| | Distance | 15 m | 0.80 | 0.18 | 0.07–0.47 | 0.37 | 0.642 |
| | Distance | 65 m | 0.14 | 0.03 | 0.00–0.59 | 0.08 | 0.185 |
| | Soak time | 4–5 days | 6.39 | 1.47 | 0.19–11.3 | 72.0 | 0.074 |
| | Soak time | 8 days | 4.98 | 1.15 | 0.11–12.3 | 61.6 | 0.185 |
| | Soak time | 12–14 days | 6.55 | 1.51 | 0.19–12.0 | 79.1 | 0.076 |
| | LF density | | 0.98 | 0.23 | 0.21–0.24 | 0.24 | 0.454 |

Effect of deployment factors and lionfish density on mean recruitment (fish observed within the trap footprint during retrieval) and catches (fish landed aboard the vessel) of lionfish and native fish. Factors examined included number of traps (single or paired), distance from the adjacent artificial reef (5 m, 15 m, or 65 m), soak time (0.25 days, 1 day, 4–5 days, or 12–14 days), and adjacent reef lionfish density (fish/100 m$^2$). Differences in means were tested with generalized linear mixed models (GLMM) fit with negative binomial error distributions and reef site as a random effect. The GLMM outputs show the log-linked parameter estimates for mean number of fish recruited or caught per trap, associated 95% confidence intervals (CI) around the parameter estimate, and the odds ratio (proportional effect). Odds ratio and hypothesis testing (z- and P-values) represent the difference from the GLMM intercept, i.e., the difference compared to a single trap deployed 5 m from the reef with a soak time of 0.25 days (recruitment model intercept) or 1 day (catch model intercept).
*P-values <0.05 are bolded.

data [54]. GLMM results indicated lionfish recruitment and catch were significantly affected by distance to the adjacent artificial reef and soak time (**Table 3**). Mean lionfish recruitment to paired traps was 56% that of single traps, although the difference was not significant (P = 0.195) due to high variance (**Fig 5A**). Traps placed at close (5 m) and intermediate (15 m)

distances had similar recruitment to each other (P = 0.935), while traps placed far (65 m) from a reef recruited significantly fewer lionfish (P < 0.001) (**Fig 5B**). Lionfish recruitment was highest for traps deployed for 1 day (**Fig 5C**). Lionfish catch was 84% lower for the longest soak time of 12–14 days (P = 0.001) and lionfish recruitment for that treatment was 39% lower

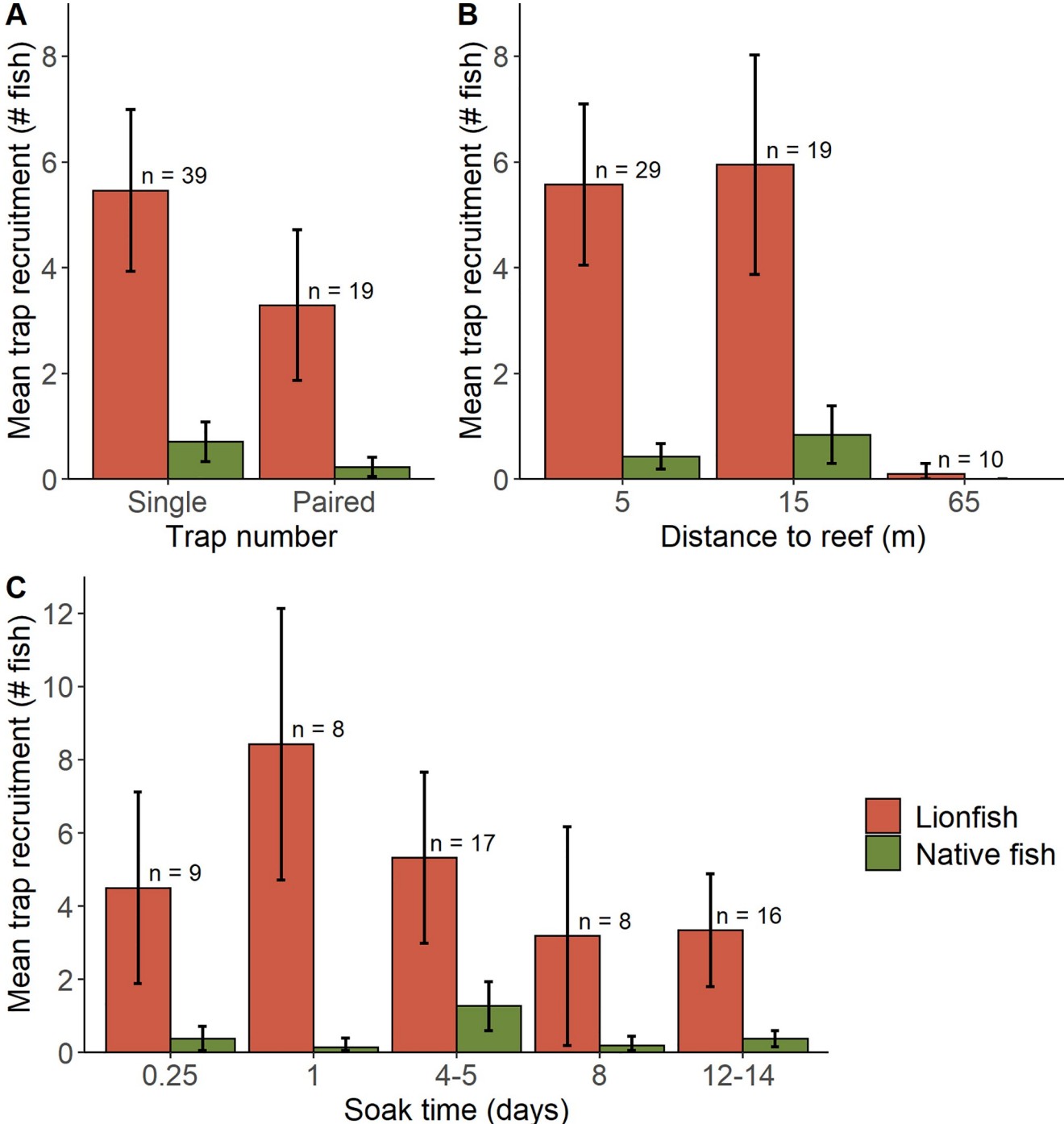

**Fig 5. Lionfish and native fish recruitment.** Mean (± 95 CI) count of lionfish and native fish observed within the trap footprint during retrieval, with the number of replicates per level indicated (n). Trap deployment configurations examined factors of A) distance to the adjacent artificial reef, B) number of traps, and C) soak time.

(**Table 3**). Lionfish density on the artificial reef study sites ranged from 6 to 36 fish per 100 m$^2$. Unexpectedly, lionfish density on the adjacent artificial reefs was not a significant covariate in predicting lionfish recruitment (P = 0.152) nor lionfish catch (P = 0.267). Mean native fish recruitment and catch were <1 fish per trap (**Fig 5**) and the GLMM results indicated native fish recruitment and catch were significantly affected by trap number and soak time (**Table 2**). Paired traps had approximately 75% lower recruitment and catch per trap than single traps (**Fig 5A**, **Table 2**). In contrast to the lionfish models where recruitment and catch were lower with longer soak times, recruitment and catch for native fishes increased during the longer soak times of 4–5 days, 8 days and 12–14 days (**Fig 5C**). Longer soak times predicted 4–12 times higher recruitment and catch of native fish, although only the 4–5 day level in the native fish recruitment model was significant (P = 0.013).

Lionfish size distributions were multi-modal for both those caught by traps (n = 137) and by spearfishing (n = 3,063) (**Fig 6**), which is likely due to distinct TL modes for juvenile and adult lionfish [52,55]. Results from the KS test indicated lionfish TL distributions were not significantly different between those captured from traps deployed at distances 5 m and 15 m (**Fig 6A**, P = 0.591). Lionfish sizes from 65 m distanced traps were not tested because only one lionfish was caught in this treatment. Probability density plots for trap-caught and spear-caught lionfish from similar artificial reefs in the same 6-month period showed that traps captured disproportionately fewer juvenile (<200 mm) and more large adult (>300 mm) lionfish than those caught via spearfishing (**Fig 6B**). Mean TL for trap-caught lionfish (277 mm) was 19 mm larger than spear-caught lionfish (296 mm) and had a significantly different distribution (KS test, D = 0.149, P = 0.006). Given the length-weight relationship for nGOM lionfish, mean weight for trap-caught fish were estimated to be 62 grams higher. There was no

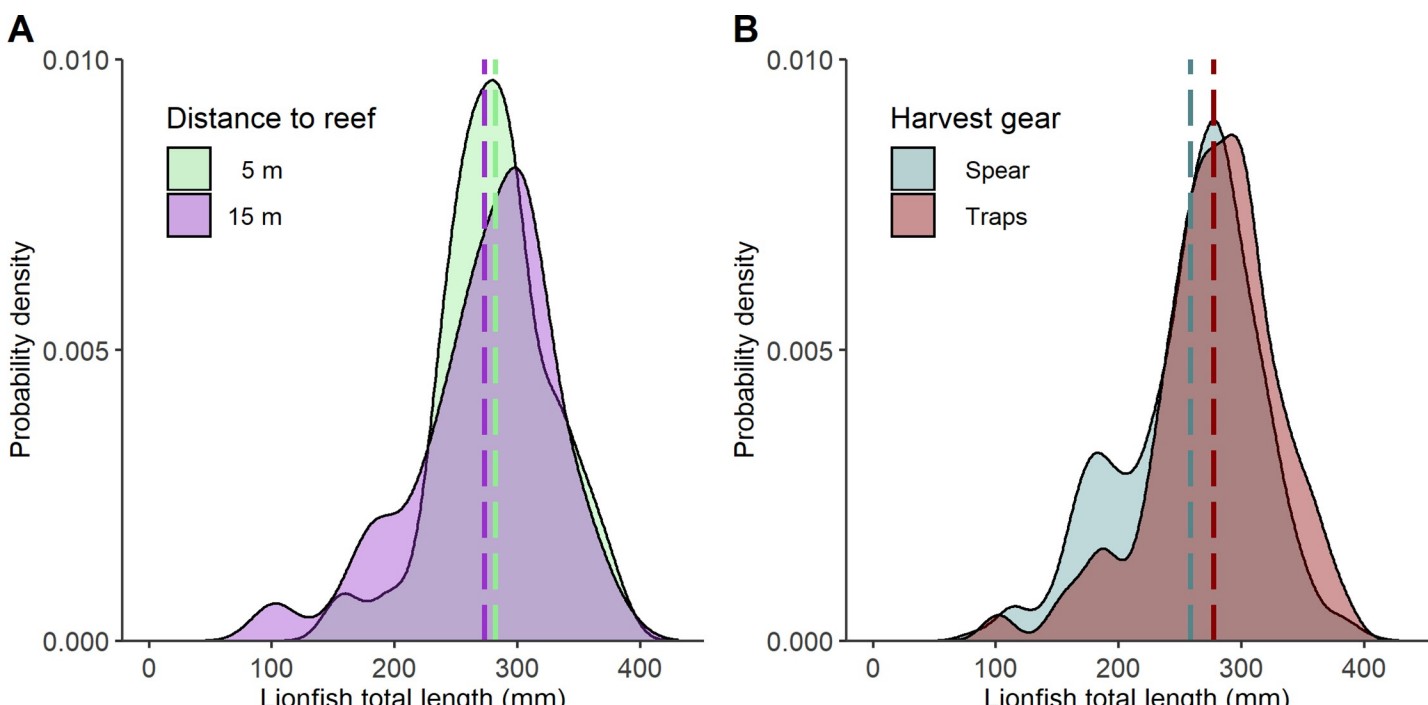

**Fig 6. Lionfish size distributions by distance to reef and harvest gear.** A) Lionfish total length (TL) probability density distributions for lionfish captured in Gittings traps deployed at 5 m and 15 m distances to the adjacent artificial reef. B) Lionfish TL distributions for lionfish harvested by Gittings traps (all treatments) and lionfish sampled concurrently (June–December 2018) by spearfishing on similar northern Gulf of Mexico artificial reefs. Mean TL per distribution is indicated by a vertical dashed line.

significant difference in the size distributions for trap-caught lionfish between seasons Jun–Sep vs. Nov–Dec (KS test, D = 0.192, P = 0.201).

## Discussion

The lack of apparent environmental impacts of Gittings traps suggests they may be suitable for further field testing. Lionfish recruitment and catch were >10X higher than that of native fish and traps did not move during severe weather events. Further testing will be needed, however, to determine the potential for movement caused by attaching surface floats. Because wide-scale commercial use of Gittings traps would entail remote deployment and recovery from the surface, design iterations and field tests will be necessary to improve deployment success and lionfish catch rates. Potential gear modifications could include the following: adjustments to flotation and ballast to help the traps maintain a vertical orientation during descent and ensure successful opening, a reconfigured harness that closes the jaws more quickly and keeps them closed during recovery, a looser net that could allow more billowing and not contact lionfish during closure, and faster trap retrieval provided by a shipboard winch. Trap mass will also be a consideration for commercial application. The traps in this study were built with relatively thick (#6) rebar and weighed approximately 35 kg, which made it difficult for a single person to move or deploy. Moreover, the need to contract an industrial rebar bender to bend this thick rebar into the semi-circular jaw frames constituted >50% of their construction costs. Either #4 (13 mm/0.5" diameter) or #5 rebar (15.875 mm/0.625" diameter) are more easily sourced and manipulated, and may be better materials for trap jaws. Recent work has also examined other design modifications for reducing production costs of Gittings traps, such as an octagon rather than a circular shape and weldless construction (pers. comm., S. Delello, ReefSave.org).

Lionfish removal efficiency by Gittings traps placed near artificial reefs was 12–26%, which is higher than many fisheries where reported removal efficiency is <10% [56–58]. A lower removal efficiency, which would leave uncaptured lionfish, will need to be considered when evaluating the potential community benefits offered by lionfish trapping. Lionfish removals do not necessarily translate into ecological benefits [14,59,60], given that lionfish predation [61,62], growth [53,63], and colonization [14,51,60] rates are controlled via density-dependent feedbacks. Ecosystem models may thus be appropriate for examining the potential community effects of a deepwater lionfish fishery [15,64]. Removal efficiency rates by Gittings traps were considerably lower than the >85% removal efficiency estimated for spearfishing lionfish on nGOM artificial reefs [46]. Considering that spearfishing can reduce lionfish densities in areas frequented by divers [65–67] and that spearfishing fisheries have caused severe depletion of other reef fishes [68–70], we expect spearfishing to remain the most efficient and cost-effective method for removing lionfish biomass at depths <40 m.

Testing different deployment configurations indicated that lionfish catch near artificial reefs is potentially optimized by deploying a single trap 15 m or less from a reef site for one day. Mean recruitment with this deployment configuration was approximately 5 lionfish and 0.1 native fish per trap. Higher lionfish recruitment to traps deployed closer to reefs was likely attributable to the high site fidelity [71] and central-place foraging [13] of lionfish. More sampling at distances between 15 m and 65 m could determine the effective attraction distance of traps and the linearity of this relationship. Given that lionfish are gregarious and often found in groups [72], and that aggregating behavior is driven by broad-scale habitat complexity [73,74], we considered the hypothesis that paired deployments synergistically recruit more lionfish. However, we found lionfish recruitment and catches were similar between single versus paired traps. Tests of alternative soak times indicated that a one-day deployment had the

highest recruitment of lionfish and lowest recruitment of native fish. Though these were not significant different from other soak times, it suggests that longer soak times may be unnecessary. It may also suggest that lionfish recruit relatively quickly to traps but gradually leave, while native fish take longer to recruit to traps but tend to stay. Our observation of an inverse correlation between lionfish and native fish abundance, particularly during the early stages of recruitment to the traps, appears consistent with findings from other studies that report indirect effects of lionfish and that their presence deters native fish from occupying reef space [75–78].

Overall lionfish recruitment numbers to the Gittings trap were about double the numbers observed on lobster traps deployed near mesophotic reefs in Bermuda [34]. Subsequent experimental tests on Bermudan reefs with modified lobster trap funnels found that, similar to this study, lionfish catches were highly variable and right-skewed: mean recruitment to the lobster trap structure was approximately 3 lionfish per trap and ranged to >15 lionfish per trap [79]. While this variation was likely driven by the spatial variability observed in lionfish densities on Bermuda reefs [9,80], our results unexpectedly indicated that lionfish density on an adjacent artificial reef was not a significant predictor of lionfish recruitment or catch by Gittings traps. We had anticipated that lionfish movement from reefs to traps would be correlated with site density, as high lionfish densities have been related to local prey depletion [12,14,81], cannibalism [82,83], lower body condition [53], and greater movement on coral reefs [61,84].

The apparent selectivity of Gittings traps for larger lionfish could have implications for depleting their populations. Smaller lionfish may have escaped from the traps at greater rates and, in retrospect, it would have been useful to collect size estimates of the lionfish by divers prior to trap retrieval. Alternatively, traps may attract larger individuals. Higher movement rates for larger lionfish is reasonable as they are more physically capable to make such movements and face lower risk of predation in transit [85,86]. Juvenile lionfish risk conspecific predation [82,87], particularly in areas of high lionfish density where this study was conducted [83], and smaller individuals may thus occupy smaller foraging areas near reefs [88]. Given fecundity increases exponentially with length [89–91], a gear selectivity bias toward larger fish may have a greater impact on reducing lionfish egg production. High enough reductions of large spawners would lead to recruitment overfishing whereby the reduced spawning stock decreases future population growth [92]. That said, removal efforts aimed to deplete lionfish biomass should also target juveniles. Harvest of faster-growing juveniles [53,55,93] contributes to growth overfishing [94], and age-structured lionfish population models indicate that controlling lionfish population growth requires removing smaller individuals [23,24].

The potential for lionfish traps or other novel harvest technologies to reduce deepwater lionfish biomass will be contingent on their capacity to harvest them from natural reefs. Natural reefs constitute over 99% of the region's reef habitat, with 90% being mesophotic reefs deeper than 40 m [95]. Even on shallower natural reefs accessible to divers, spearfishing catch rates are limited by lower lionfish density [3,53] and lower removal efficiency [46]. Similar gear testing to that conducted in this study will be needed for traps deployed near natural reefs to evaluate their potential for habitat damage or bycatch, considering that these reefs have differences in benthic structure and community composition [14]. Remote recovery of traps may also influence bycatch rates and species composition as some native fish could have reacted to divers and left the trap footprint before it closed. Ultimately, it will be critical to determine how lionfish attraction to traps deployed near natural reefs compares to the recruitment and catch rates observed in this study. Lionfish densities on nGOM artificial reefs are >10X higher than on nGOM natural reefs [3,46,53], which suggests a trap structure may readily attract lionfish from natural reefs. However, if lionfish movement and home range are largely driven by density-dependence or food availability [14,61,84,96], then the substantially lower densities on

natural reefs may critically reduce catch rates. Therefore, the potential economic viability of a lionfish trap fishery should be carefully considered. Future trap field testing should examine design strategies to increase lionfish attraction (e.g., using lights, sound, or different structures) and technoeconomic assessments may identify capital and operational expenses to determine what catch rates could make lionfish trapping commercially feasible.

## Supporting information

**S1 File.**
(R)

**S2 File.**
(R)

**S3 File.**
(CSV)

**S4 File.**
(CSV)

**S5 File.**
(CSV)

**S1 Data.**
(ZIP)

**S2 Data.**
(ZIP)

## Acknowledgments

We thank Josh and Joe Livingston (DreadKnot Charters), Kara Wall (Florida Fish and Wildlife Research Institute), Tony Reyer (NOAA), Sal DeLello (ReefSave.org), Laura Tiu (Florida Sea Grant), Stacy Frank (Lionfish University), Jim Hart (Lionfish University), Alexandria Tucker (University of Florida), Dominic Andradi-Brown (University of Oxford), Kelli O'Donnell (NOAA), and Florida Fish and Wildlife Conservation Commission personnel Alan Pierce, Kali Spurgin, Amy Brower, Hanna Tillotson, and Michael Kennison.

## Author Contributions

**Conceptualization:** Holden E. Harris, Alexander Q. Fogg, Stephen R. Gittings.

**Data curation:** Holden E. Harris, Alexander Q. Fogg.

**Formal analysis:** Holden E. Harris, Robert N. M. Ahrens, William F. Patterson III.

**Funding acquisition:** Holden E. Harris, Alexander Q. Fogg, Stephen R. Gittings, Robert N. M. Ahrens.

**Investigation:** Holden E. Harris, Alexander Q. Fogg, Stephen R. Gittings, Robert N. M. Ahrens.

**Methodology:** Holden E. Harris, Alexander Q. Fogg, Stephen R. Gittings, Robert N. M. Ahrens.

**Project administration:** Holden E. Harris, Stephen R. Gittings, Robert N. M. Ahrens, Micheal S. Allen, William F. Patterson III.

**Resources:** Holden E. Harris, Stephen R. Gittings, Micheal S. Allen.

**Software:** Holden E. Harris.

**Supervision:** Holden E. Harris, Stephen R. Gittings, Robert N. M. Ahrens, Micheal S. Allen, William F. Patterson III.

**Validation:** Holden E. Harris.

**Visualization:** Holden E. Harris.

**Writing – original draft:** Holden E. Harris, William F. Patterson III.

**Writing – review & editing:** Holden E. Harris, Alexander Q. Fogg, Stephen R. Gittings, Robert N. M. Ahrens, Micheal S. Allen, William F. Patterson III.

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
