## [Decision Letter · Decision Letter 0]

5 May 2020

PONE-D-20-07219

Testing the efficacy of lionfish traps in the northern Gulf of Mexico

PLOS ONE

Dear Mr. Harris,

Thank you for submitting your manuscript to PLOS ONE. After careful consideration, we feel that it has merit but does not fully meet PLOS ONE’s publication criteria as it currently stands. Therefore, we invite you to submit a revised version of the manuscript that addresses the points raised during the review process.

More specifically, along with the (many) minor and editorial comments the authors should address the following issues. In discussion the significance of the findings of the manuscript should not be overstated. The size effect should be discussed along with the behavior and some details on the size of native fish. Some details on the density of the lionfish populations should be provided. 

We would appreciate receiving your revised manuscript by Jun 19 2020 11:59PM. To enhance the reproducibility of your results, we recommend that if applicable you deposit your laboratory protocols in protocols.io, where a protocol can be assigned its own identifier (DOI) such that it can be cited independently in the future. For instructions see: http://journals.plos.org/plosone/s/submission-guidelines#loc-laboratory-protocols

We look forward to receiving your revised manuscript.

Kind regards,

Athanassios C. Tsikliras

Academic Editor

PLOS ONE

Journal Requirements:

Reviewers' comments:

Reviewer's Responses to Questions

**Comments to the Author**

1. Is the manuscript technically sound, and do the data support the conclusions?

Reviewer #1: Yes

Reviewer #2: Yes

2. Has the statistical analysis been performed appropriately and rigorously? 

Reviewer #1: Yes

Reviewer #2: Yes

3. Have the authors made all data underlying the findings in their manuscript fully available?

Reviewer #1: Yes

Reviewer #2: Yes

4. Is the manuscript presented in an intelligible fashion and written in standard English?

Reviewer #1: Yes

Reviewer #2: Yes

5. Review Comments to the Author

Reviewer #1: This is an interesting manuscript that is very applied in focus, testing the effectiveness of the Gittings lionfish trap. It is a generally well written and concise manuscript that provides much useful information in thinking about approaches to managing lionfish populations in areas that are beyond the depth range of recreations SCUBA diving. I have a few broader comments and some suggestions for improvement, but do not think there is much work to be done for this to be in an acceptable form for publication.

Broader comments:

Methods:

I think we need a little more info in the methods about the artificial reefs. For example, how large are each of these structures. What’s the dominant habitat type that has established on them. What’s their relief relative to the seabed. What is the composition of the seabed around the structures that the lionfish will have to swim over to get to the traps. What’s the general water visibility typically like there - i.e. are traps in visual distance for fish from the structure. Does the point count for fish done by the diver in a 15m wide cylinder fully encompass the whole of each artificial reef?

L121 and L126 - need to clarify why these n numbers are different. Also define what ‘paired’ means - are these two traps deployed near each other at the same distance from the artificial reef with the same soak time? What was the distance between paired traps. Could be worth adding a image of pairs traps to Figure 1 or Figure 3.

The breakdown of number of replicates between each of the factors is unclear (i.e. how many traps for each combination of number of traps, distance to reef, and soak time. Please add a summary table here that shows the number of replicates for each combination.

Please add a lot more detail on where the spearfishing data came from. Is this a scientific monitoring/lionfish culling dataset? Or does it come from recreation divers and lionfish derbies? Please elaborate on how fishers were able to capture all lionfish regardless of size - note the papers that suggest visual survey methods for lionfish/spearfishing are likely to be biased towards larger individuals anyway: e.g. https://doi.org/10.1007/s00338-012-0987-8

Also, there is evidence of recruitment peaks in lionfish populations. Was the spearfishing data collected over a similar time period to the trap data? Note, there is seasonality in size distribution of lionfish in some regions which may also affect the comparability of this data if not collected at the same time of year. e.g. https://doi.org/10.7717/peerj.2730 If this is a problem, I think the comparisons between collection methods can still be included, but would note this issue in the discussion.

Results:

Table 2 - Why is a soak time of 1 day set as the intercept? This confused me when I first looked at it, as the other other variables all use the smallest unit (i.e. single trap, 5 m distance from reef) as the intercept to compare others levels of those factors to. I suggest using the 3-6 hours soak time as the intercept to compare the 1 day, 4-5 days, and 12-14 days soak times to.

I know this is not directly a question you set out to address, so please feel free to ignore. However, it would be very interesting to know whether there is a relationship between size of lionfish caught in the trap and distance from reef the trap was. There’s a lot of literature on ontogenetic fish movements that would suggest that traps further from the reef would be more likely to be biased towards larger lionfish. Maybe add as a second panel to Fig 6?

Discussion:

There’s some numbers on bycatch from studies of lobster traps for lionfish - e.g. Pitt and Trott 2015, though the authors may be aware of other studies as well. It would be good to place the report numbers in L250 in the context of these other studies.

Pitt JM, Trott TM (2015) Trapping lionfish in Bermuda, part II: lessons learned to date. In: Proceedings of the 67th Gulf and Caribbean Fisheries Institute, Christ Church, Barbados, 3–7 November 2014, pp 221–224

General comments:

Fig 2 - Add scale bars and north arrows to all parts of the figure. The boundaries of the green study area box in the lower right panel seems to be positioned further south than the area indicated in the upper panel. I would suggest labeling each panel A, B, C and then providing a brief description of each panel in the legend.

This manuscript makes extensive references to technical reports and videos. I think the nature of this study requires this, however, for long-term access web links to government sites etc are unlikely to be stable. I notice many of them are by Gittings who is a coauthor. Please could these technical reports and videos be cited via a DOI to ensure they are accessible to readers longer term?

Minor comments:

L18 - add by SCUBA - so that it reads: Spearfishing by SCUBA is currently…

L25 - be clearer what is meant by regulated species

L49-50 - add word ‘invasive’ so that it reads: However, invasive lionfish have been…

L52 - mesophotic is defined from 40 - 200 m depth. So edit sentence to say: ‘lies within mesophotic and upper-bathyal depths of 40-300 m’

L53 - also see this recent review of mesophotic and upper-bathyal lionfish and their potential impacts in the Western Atlantic that may be useful reference to cite here:

https://doi.org/10.1007/978-3-319-92735-0_48

L71 - change to: ‘However, it is critical’…

L96 onward - add some labels to Fig 1 to point out the key elements that are being described in the text. e.g. the trap jaws, the pivot axel, the netting, the lattice etc

L111 - show location of Destin Fig 2

L130 - can you be more clear here that you are defining a variable called ‘Lionfish catch’ that you will be reporting on from the models. And also be more clear about defining “lionfish recruitment’ it wasn’t clear to me on my first read that you were defining terms here, and then I was confused when I got to the results how these terms were defined.

L133 - ‘adjacent reefs’ - why is this plural? You mean the lionfish density on the single artificial reef next to the trap? Please rework this sentence to improve clarity

L176-177 - This sentence on trap bycatch is very confusing. I know you explain what you mean in later sentences with the undetected individual, but I would remind readers upfront that ‘recruiting’ means diver in water observations and then give that result, and then remind readers that ‘’caught’ means identified on the surface in the trap after the dive and give the caught result. Then state that clearly there was a detection error.

L224 - these numbers are per trap right, not in sum for the pair? Please clarify

L267 - I think you need to discuss that these results represent diver collected traps. So the intention is that these can longer-term be remotely deployed and recovered from the surface by boat, so need to trial in these conditions/verify that these results hold up with a boat recovery.

L274 - also weight is a big consideration when deploying to much deeper reefs, i.e. winch for boats etc for recovery. As you say in your intro, there is a need for a method that could allow capture down to >300 m!

L288 - though trapping removes worries around health and safety of divers in the water, and can have multiple simultaneous traps deployed

L292 - See the work of Green et al 2014 that specifically identifies target thresholds for culling to translate to ecological benefits. There is a need to calculate these thresholds for mesophotic reefs, and then consider whether traps can reach them: https://doi.org/10.1890/13-0979.1

At some places in the manuscript you refer to people who fish as ‘fishers’ in others you use ‘fishermen’. I would recommend standardizing to fishers, or be clearer why there’s a need for the two terms.

Reviewer #2: Overall the manuscript was clearly and concisely written and provides valuable information about an important marine invasive species of concern throughout the Western Atlantic. The research of the efficacy of utilizing a non-containment trapping device to capture lionfish near artificial reefs is important for improving management strategies regionally to control lionfish populations on mesophotic reefs. The paper highlights the potential of these traps to capture lionfish and a produces a capture rate comparable to trapping of other native target species using fish traps. It discusses recommendations for future directions of research to help improve upon this potential methodology by reducing escape rates of lionfish and placing traps at optimal distance to reefs.

The manuscript is recommended for acceptance with minor revisions described below.

Major Issues

1. The author should be careful to not overstate findings when discussing the significance of larger size of lionfish in traps. There are a few causes of this result that were not addressed and should be if possible (month(s) which most lionfish were capture from traps compared to spear, is there size estimates of all recruits from diver surveys that can be looked at to see if only larger fish recruited to traps. Did more small fish escape the traps.

2. The author fails to discuss the fact that no native fish escaped from the trap. This is relevant information since upwards of 50% of lionfish escaped. Authors should provide a better description of the native fish behavior observed in the traps during this study which could help with improvement to future design.

3. Authors failed to describe what happened to the native fish that were captured in the traps? Were they descended and was that successful (ie did the bycatch survive)? Did the authors estimate the size of these fish? Were they on par with the sizes of lionfish in the traps?

4. The author should provide more detailed information about the density of the lionfish populations at each site. Without this information it will be difficult for future studies assessing efficacy of capture methods to compare their results to this current study. Will these traps work at all on sites with much lower lionfish densities?

Minor issues

Page: 8

Line 25: Is this species or individual fish

Line 25: which "are" regulated

Line 26: Here you have 29 fish captured which is higher than the 28 recruited unless the former is species not individual fish. These sentences are a bit confusing in the abstract. More clearly explained in results.

Line 29: Is this captured or recruited? Can you get estimates on recruited from videos? Would be interesting to know if the smaller fish are more likely to escape or just less likely to recruit to the trap structure. Need to set up some stereocameras probably to get at that question.

Page 9

Line 48 "is a top"

Line 53 - This may be an overstatement at the knowledge of population densities at mesophotic depths throughout the western Atlantic since site specific information is very limited to date.

Page 10

Line 66: Should this be Figure as well if you are introducing a descriptive table, for a ready that has not seen the trap a figure would have more use at this time?

Line 71: "economic viability" and potential undesirable effects.

Page 11

Line 85 - This is different that what was reported in abstract. Compared to spearfishing catch

Page 12

Line 110 - Four of these sites look quite close together compared to the other sites. Did you account for distance to next structure when analyzing recruitment or capture results or lionfish densities at site?

Did you look at artificial reef type when analyzing recruitment results or lionfish densities at site?

Line 116 - Did you estimate size on these surveys as the in situ size estimates referred to above?

Page 13

Line 127 - Need to explain why traps were retrieved by divers and not set as a surface buoy. This would not be a practical way to deploy and retrieve these traps commercially.

Line 130 - Was there a size estimate conducted here by the diver?

Line 135 - Same comment as above...

Four of these sites look quite close together compared to the other sites. Did you account for distance to next structure when analyzing recruitment or capture results or lionfish densities at site?

Did you look at artificial reef type when analyzing recruitment results or lionfish densities at site?

Page 14

Line 145 - was this done at same time period as the trap study? Within a week/month/year? That could greatly influence size distribution.

Line 150

What did you do with the native species captured? Did you descend them? Assess condition upon release and mortality?

Line 152 - Was there any issues with trap entanglement with habitat?

Page 15

Line 177 – Ok this is clear here but this detail makes the abstract a bit confusing without it being explained.

Line 170 – 183 - I would reword this as "Four native species capture in the traps were regulated species:....

Then

"Addition native species catches consisted of 15 sand perch....

Line 183

There is no discussion at all about why all native fish that recruited to the traps were captured. Did any of them try to escape as it was being retrieved? This would be worth discussing to potentially brainstorm why lionfish are more likely to escape and what modifications can be made to take into account their different behavior.

Did a high proportion of lionfish escape when their were more total fish in the trap before retrieval?

Page 16

Line 189 - remove "to"

Line 192 - Since this influences catch would be interesting to look at effects of having multiple artificial reefs nearby.

Line 193 - But did the two traps catch more lionfish in total than one? Because that would suggest it might still worth putting two traps down on a site, particular if they are connected and you can deploy and retrieve them with minimal additional effort than two.

Line 199 - Did you have video data to see if lionfish had recruited to the traps within the 1-5 days when recruitment is highest but then left before you retrieved it day 12-14?

Line 202 - How low was the lowest density. Is this because density was just relatively high at all sites?

Page 18

Line 235

As commented above. Were the fish speared during the same time period (within the same month at least) because if not this could explain less juveniles.

Did you look at videos or estimates during surveys to see if smaller fish escape more easily or are they less likely to recruit?

Page 19

Line 242 - I see the time period here. It would be worth checking if the portion of fish caught in the summer months in traps is similar to the fish caught in the summer months spearing. Since the trap catch sample size is relatively low (especially compared to spearing sample size); if a larger proportion of fish were caught in winter just this could explain less small fish.

Line 251 - Change Would be expected to "was expected"

Line 256 – Is there any evidence of them leaving? Camera footage?

Line 258 - But lionfish were still present during the long soak times no just slightly lower densities? This suggests that native fish may just take longer to recruit not that lionfish necessarily deter them. Unless you compared traps with and without lionfish with native predators densities, but I didn't see that in the results.

Page 20

Line 261 - As stated above, it would be good to know whether the total number of lionfish for the two traps was larger or similar to that of one trap and not just the per trap density. It seems from the table that it is likely the same.

Line 263 - As state above in Results.

What type of densities are we comparing here? That would be useful for the reader to know to understand whether artificial reefs may be at carrying capacity or etc.

Were they all relatively high density?

Line 271 - Wouldn't this loose net increase risk of entangement and ghost fishing

Page 21

Line 297 - You don't know it is higher movement/recruitment of large individuals to the traps. Maybe the small fish escaped the trap more easily and did not get captured. Be careful not to overstate a result here about smaller fish movement.

Line 306 - This sentence (Harvest…) is confusing, not sure exactly what you are saying here

Page 22

Line 318 - But that is because there is no other structure for them to attract to which is not the case as natural reefs. I don't think we can assume traps will readily pull lionfish from natural reefs.

Figure 1C – what is that top line attached to in the schematic? Above the float?

Figure 6 – Density on the y axis is confusing here since it is used in earlier figures and tables to represent density of fish per trap

6. PLOS authors have the option to publish the peer review history of their article (what does this mean?). If published, this will include your full peer review and any attached files.

Reviewer #1: Yes: Dominic A Andradi-Brown

Reviewer #2: No

---

## [Author Response · Author response to Decision Letter 0]

29 Jun 2020

Dear editor,

The manuscript was revised considerably based on the edits and comments from the referees and editor. Addressing these comments and concerns improved the scientific rigor, clarity, and overall presentation of the study. Please see the revised manuscript uploaded (with and without tracked changes). Specifically, our revision provides details and clarification based on the reviewer guidance for the following: describing the study sites, defining lionfish population densities per site, showing the replicates in the statistical analysis, describing observations of fish behavior, discussing detection error for native reef fish species, clarifying the timing for spearfishing sampling, and discussing other lionfish trap research. Furthermore, we better acknowledge our data limitations, the high variance in some of the results, and walk-back the strength of some of the conclusions made in the discussion (e.g., the difference in fish size). Our purpose in this study was to show our work testing an innovative harvest gear for invasive lionfish, consider the potential limitations and applications based on our findings, and discuss directions of continued research and development. Addressing the reviewer comments helped to better achieve these objectives and we believe publishing this revised article in PLOS ONE will make a valuable scientific contribution.

The 'Response to reviewers' document details our point-by-point responses to the questions, issues, and requested changes by the reviewers. We were able to make nearly all of the requested changes or clarifications from the referees. Any questions or changes that could not be answered are explained or defended. We think the editor will find our revised article suited (or nearly suitable) for publication, and we are happy to make any further changes or address any further issues. 

Thank you for your consideration,

HEH, AQF, RNMA, MSA, and WFP

27 June 2020

---

## [Decision Letter · Decision Letter 1]

20 Jul 2020

PONE-D-20-07219R1

Testing the efficacy of lionfish traps in the northern Gulf of Mexico

PLOS ONE

Dear Dr. Harris,

Thank you for submitting your manuscript to PLOS ONE. After careful consideration, we feel that it has merit but does not fully meet PLOS ONE’s publication criteria as it currently stands. Therefore, we invite you to submit a revised version of the manuscript that addresses the three minor points raised by one of the reviewers.

We look forward to receiving your revised manuscript.

Kind regards,

Athanassios C. Tsikliras

Academic Editor

PLOS ONE

Reviewers' comments:

Reviewer's Responses to Questions

**Comments to the Author**

1. If the authors have adequately addressed your comments raised in a previous round of review and you feel that this manuscript is now acceptable for publication, you may indicate that here to bypass the “Comments to the Author” section, enter your conflict of interest statement in the “Confidential to Editor” section, and submit your "Accept" recommendation.

Reviewer #1: All comments have been addressed

Reviewer #2: All comments have been addressed

2. Is the manuscript technically sound, and do the data support the conclusions?

Reviewer #1: Yes

Reviewer #2: Yes

3. Has the statistical analysis been performed appropriately and rigorously? 

Reviewer #1: Yes

Reviewer #2: Yes

4. Have the authors made all data underlying the findings in their manuscript fully available?

Reviewer #1: Yes

Reviewer #2: Yes

5. Is the manuscript presented in an intelligible fashion and written in standard English?

Reviewer #1: Yes

Reviewer #2: Yes

6. Review Comments to the Author

Reviewer #1: Thank you for comprehensively addressing my previous review comments and updating the manuscript. I have no further comments.

Reviewer #2: The author has done a thorough job of addressing the comments and suggestions by both reviewers and improving the clarity of the methodology and results and expanding the discussion of the manuscript. It is an interesting, focused and well-written manuscript and with very minor revisions it will be acceptable for publication.

Minor comments

Line 105 remove space before “,”

Line 219 – This sentence as stated appears unfinished as if you are not going to provide more details since it is the end of the section. May add “as described below.” to the end so the reader knows there is more coming in the next section.

Native Fish Recruitment:

Have you considered that the presence of a SCUBA divers in the water prior to retrieval may have influenced the amount on native fish that were observed “recruitment”.

Certain native species would be more likely to swim away as divers approach compared to lionfish (which are relatively unafraid) and therefore never observed by the divers. These individuals may therefore be at risk of being captured by the trap as bycatch if retrieved by the boat without divers. It could be argued that fish skittish of divers would also be skittish of a moving trap and would have likely escaped as it closed. This may be worth discussing briefly.

285-290

The authors have addressed the size issue and edited the results and discussion thoroughly so no addition changes are required.

However, it would be interesting to look at the size distribution of lionfish from the spearfish each month or each season to see if there was larger number of recruiting juveniles captured in June and July by spear and therefore skewing this comparison? Realizing the sample size from the traps is likely not large enough to run this comparison per month but maybe seasonally? June-August and September-December?

Line 344

As you stated in the sentence above soak time may be the factor in the density of native fish not lionfish presence. Careful not to overstate the effect of lionfish presence on native species density. Maybe it was the opposite and the later recruitment of native species chased the lionfish away?

7. PLOS authors have the option to publish the peer review history of their article (what does this mean?). If published, this will include your full peer review and any attached files.

Reviewer #1: **Yes: **Dominic A Andradi-Brown

Reviewer #2: No

---

## [Author Response · Author response to Decision Letter 1]

25 Jul 2020

This revision addressed several minor changes requested by Reviewer 2. All requested changes were made, and they improved the manuscript. The most substantial change was the addition of a test to examine whether season could have affected lionfish size distributions, as described below. 

We also took this opportunity to make minor changes to language and syntax to help clarify the material. Although we do not consider any changes substantive, we’d like to point out three in particular to the editor: 

1. The term “native species” was changed throughout the manuscript and Fig. 5 to “native fish”. This clarifies the term to represent the number of individual fish, rather than the number of different species: e.g., a mean catch of X lionfish and Y native fish (rather than “native species”) per trap. 

2. In Table 3, the column heading “Mean count” was changed to “Parameter estimate (#fish / trap)”. This clarifies that the number given is the mean estimated from the GLMM, rather than the observed mean.

3. One citation was also added [96] for a paper that was recently published (lines 392-393): “. . . if lionfish movement and home range are largely driven by density-dependence or food availability [14,61,84,96] . . .”

Further details of our response to reviewer comments are provided in the updated and attached "Response to Reviewers" document.

---

## [Editor Report · Decision Letter 2]

29 Jul 2020

Testing the efficacy of lionfish traps in the northern Gulf of Mexico

PONE-D-20-07219R2

Dear Dr. Harris,

We’re pleased to inform you that your manuscript has been judged scientifically suitable for publication and will be formally accepted for publication once it meets all outstanding technical requirements.

Kind regards,

Athanassios C. Tsikliras

Academic Editor

PLOS ONE
---

## [Editor Report · Acceptance letter]

3 Aug 2020

PONE-D-20-07219R2 

Testing the efficacy of lionfish traps in the northern Gulf of Mexico 

Dear Dr. Harris:

I'm pleased to inform you that your manuscript has been deemed suitable for publication in PLOS ONE. Congratulations! Your manuscript is now with our production department. 

Kind regards, 

on behalf of

Prof Athanassios C. Tsikliras 

Academic Editor

PLOS ONE